# A Green Synthesis Strategy for Cobalt Phosphide Deposited on N, P Co-Doped Graphene for Efficient Hydrogen Evolution

**DOI:** 10.3390/ma16186119

**Published:** 2023-09-07

**Authors:** Jingwen Ma, Jun Wang, Junbin Li, Ying Tian, Tianai Zhang

**Affiliations:** 1School of Chemical and Environmental Engineering, China University of Mining and Technology (Beijing), Beijing 100083, China; lijunbin3293@163.com (J.L.); hlj5250@126.com (Y.T.); zta5540@163.com (T.Z.); 2PetroChina Planning and Engineering Institute, Beijing 100083, China; wangjun87@petrochina.com.cn

**Keywords:** cobalt phosphide, hydrogen evolution reaction, DNA, phosphorus source

## Abstract

The exploitation of electrocatalysts with high activity and durability for the hydrogen evolution reaction is significant but also challenging for future energy systems. Transition metal phosphides (TMPs) have attracted a lot of attention due to their effective activity for the hydrogen evolution reaction, but the complicated preparation of metal phosphides remains a bottleneck. In this study, a green fabrication method is designed and proposed to construct N, P co-doped graphene (NPG)-supported cobalt phosphide (Co_2_P) nanoparticles by using DNA as both N and P sources. Thanks to the synergistic effect of NPG and Co_2_P, the Co_2_P/NPG shows effective activity with a small overpotential of 144 mV and a low Tafel slope of 72 mV dec^−1^ for the hydrogen evolution reaction. This study describes a successful green synthesis strategy for the preparation of high-performance TMPs.

## 1. Introduction

Hydrogen energy is regarded as the disruptive technological direction of the future energy revolution. The hydrogen energy industry is now essential to the national energy strategies of many countries. The question of how to obtain hydrogen in a large, cheap, convenient, and green way is the primary problem facing the development of hydrogen energy. Water electrolysis is considered to be a promising direction for future hydrogen production, especially electrolyzing water through using renewable energy and abandoned electricity [1,2,3]. Highly efficient catalysts are crucial for reducing the energy barrier for the hydrogen evolution reaction (HER). It is well-known that, at present, platinum-based catalysts are the most effective catalysts for the HER. However, their commercial application is restricted by their expensive cost and insufficient supply. Therefore, the development of non-noble metal-based electrocatalysts with high activity and stability is urgent for practical applications.

Recently, various non-noble metal electrocatalysts, such as transition metal sulfides [4,5], nitrides [6], and phosphides [7], have been developed for the hydrogen evolution reaction. Among them, transition metal phosphides have attracted a lot of attention due to abundant reserves and superior catalytic performance in the hydrogen evolution reaction [8,9,10]. Due to the various forms of cobalt phosphides, such as CoP, Co_2_P, CoP_2_, CoP_3_, etc., the reported synthesis methods are relatively limited. Meanwhile, the commonly used method for synthesizing phosphides is the gas–solid strategy, and the main precursor of the phosphorus used is sodium hypophosphite, which may release highly toxic phosphine (PH_3_) [11,12]. Graphene has an extremely large specific surface area and excellent conductivity, making it a rational carrier for hydrogen evolution catalysts [13,14,15]. In recent years, it has been found that doped graphene can activate adjacent carbon atoms to promote hydrogen evolution due to the difference in electronegativity and size between the doped heteroatoms and carbon atoms. Qiao et al. [16] obtained N, Co co-doped graphene by using melamine as a nitrogen source and triphenylphosphine as a phosphorus source and calcining it in an argon atmosphere. Nitrogen and phosphorus heteroatoms affect the valence orbital energy levels of graphene, resulting in better catalytic activity for the hydrogen evolution by affecting nearby carbon atoms on the inactive graphene. The reaction steps for the preparation of metal phosphides and N, P co-doped graphene are generally complicated and usually include a high-temperature carbonization and a phosphating process. Therefore, it is necessary to find a simple and feasible method that can obtain NPG and directly synthesize cobalt phosphide at the same time to simplify the reaction. Deoxyribonucleic acid (DNA) is a biomolecule composed of four types of deoxyribonucleotides. Deoxyribonucleotides generally consist of one molecule of a nitrogenous base, one molecule of deoxyribose, and one molecule of phosphate [17]. From their composition, they can provide nitrogen and phosphorus for composite materials [18,19]. 

In this study, we describe the synthesis of a N, P co-doped graphene-supported cobalt phosphide composite (Co_2_P/NPG), which was synthesized through a hydrothermal reaction and a subsequent carbonization process. The biomolecule DNA was the nitrogen and phosphorus source for the preparation of NPG and Co_2_P, also acting as the chelating sites for metal ions. The as-prepared Co_2_P/NPG shows superior activity for the hydrogen evolution reaction.

## 2. Materials and Methods

Preparation of Co_2_P/NPG: First, graphene oxide (GO) was prepared using Hummers’ method. For this, 20 mg GO and 30 mL deionized water were added into a 100 mL flask; this was followed by ultrasonication for 30 min to obtain a dilute solution of GO. Then, 100 mg deoxyribonucleic acid (DNA) was added to the GO solution, and the mixture was ultrasonicated for two hours to obtain a uniformly dispersed solution. Then, the above-mentioned mixture was heated to 95 °C in an oil bath and maintained for 30 min via magnetic stirring to obtain DNA-modified GO (DNA-GO). A total of 100 mg cobalt acetate hexahydrate was dissolved in the DNA-GO solution, and 0.25 mL ammonia solution was added drop by drop into the above-mentioned mixture. After continuous magnetic stirring for 3 h, the solution was transferred into a stainless-steel reactor and maintained at 180 °C in a vacuum drying oven for 3 h. The intermediate product (Co_3_O_4_/DNA-GO) was obtained after being centrifuged and washed several times. Finally, the Co_3_O_4_/DNA-GO was placed in a quartz boat in a tubular furnace and heated to 800 °C at a heating rate of 5 °C min^−1^ in an Ar atmosphere before being kept for 2 h. After washing and filtering with deionized water, the Co_2_P/NPG composite was obtained by vacuum drying overnight. For comparison, samples with different cobalt acetate contents (50 mg and 200 mg) were also synthesized (labeled as Co_2_P/NPG-2 and Co_2_P/NPG-3).

Electrochemical measurements: All electrochemical experiments were performed on the CHI760E electrochemical workstation. Specifically, Co_2_P/NPG was used as the working electrode; a Ag/AgCl electrode and a graphite rod were used as the reference electrode and the counter electrode, respectively. In 0.5 M H_2_SO_4_, the linear scanning voltammetry curve was recorded at a scanning speed of 5 mV s^−1^. Electrochemical stability measurements were performed via 1000 CV cycles.

## 3. Results and Discussion

The fabrication process of Co_2_P/NPG is shown in Figure 1. Firstly, DNA-modified GO (DNA-GO) is coupled through the π–π stacking between DNA and GO. Negatively charged phosphate groups on DNA could provide binding sites for Co^2+^. During the preparation process, the Co^2+^ can bind to the DNA-GO due to the strong electrostatic interaction between the metal cations and the negatively charged phosphate groups on DNA. After a hydrothermal reaction, Co_3_O_4_/DNA-GO was obtained. Sufficient N and P elements in DNA enable the fabrication of Co_2_P and NPG without the need for an additional N and P source. Therefore, after carbonization, a Co_2_P/NPG composite is fabricated.

X-ray powder diffraction (XRD) was utilized to analyze the crystalline structure of the synthesized Co_2_P/NPG. As depicted in Figure 2, the red vertical line is the standard card for Co_2_P (PDF#32-0306). The XRD pattern showed five characteristic peaks located at 40.7°, 41°, 43.3°, 44.1°, 51.5°, and 52°, corresponding to the (121), (201), (211), (130), (131), and (002) crystal planes of Co_2_P, respectively. There are no peaks of metallic cobalt, cobalt oxide, or other forms of cobalt phosphide (CoP, CoP_2_, etc.), thus proving the high purity of our product. The control sample of Co_2_P was also detected via XRD, as shown in Appendix A, confirming the successful preparation of Co_2_P.

Transmission electron microscopy (TEM) was utilized to detect the morphology of the as-prepared Co_2_P/NPG. As clearly shown in Figure 3a, the Co_2_P nanoparticles are uniformly dispersed on the graphene sheet. The high-resolution TEM (HRTEM) image of Co_2_P nanoparticles (Figure 3b) presents clear lattice planes of 0.22 nm, which corresponded to the (121) plane of Co_2_P [20,21,22]. It is worth noting that the sample underwent long-term ultrasonication during the preparation process of the TEM sample, but no free cobalt phosphide particles were found during the testing process, thus proving the close binding between cobalt phosphide and N, P co-doped graphene. The energy-dispersive X-ray spectrum (EDS) results shown in Figure 3c in the region further illustrate the presence of C, N, O, Co, and P elements. The atomic ratio of Co and P derived from the EDS result was approximately 1.4:1, which is lower than the stoichiometric ratio of Co_2_P, indicating the successful doping of P onto the graphene sheet. In order to further verify the distribution of C, N, Co, and P elements in the composite, HAADF-STEM element mapping analysis was employed. Figure 3d is a STEM image of Co_2_P/NPG, and the distribution of C, N, Co, and P elements was analyzed based on this. As shown in Figure 3e,f, it can be found that the distribution of N and C elements is consistent, indicating that nitrogen was successfully doped onto graphene. The distribution of Co and P elements (Figure 3g,h) on the particles is basically consistent, which proves the successful preparation of cobalt phosphide nanoparticles.

The surface elemental composition and chemical states of Co_2_P/NPG were detected through using X-ray photoelectron spectroscopy (XPS) characterization technology. Firstly, the chemical states of the elements for N, P co-doped carbon (NPC), which was synthesized via the carbonization of DNA, were detected through XPS. The results shown in Appendix A confirm the successful preparation of NPC. As displayed in Figure 4a, the peaks can be divided into four peaks at 284.7, 286, 287.1, and 289.3 eV, contributing to the C-C, C-P, C-N, and C-O, respectively. The high-resolution spectrum of N 1s (Figure 4b) can be deconvoluted into two peaks, which can be ascribed to pyridinic N (398.2 eV) and graphitic N (400.6 eV), indicating the successful doping of N to the graphene sheet [23]. Figure 4c is the high-resolution spectrum of Co 2p, in which two peaks at 778.3 and 793.4 eV can be ascribed to Co 2p_1/2_ and 2p_3/2_ in Co_2_P [24]. The peaks at 781.2 and 797.2 eV may be due to the oxidation of cobalt ions on the surface of cobalt phosphide. The peaks at 786.0 and 802.9 eV are the satellite peaks, which can be found in transition metal-based materials. Figure 4d shows the high-resolution pattern of P 2p, which can be deconvoluted into four peaks: the two peaks at 129.5 and 130.3 eV corresponding to the P 2p_3/2_ and P 2p_1/2_ in Co_2_P, and the other two peaks located at 132.3 and 133.2 eV can be ascribed to P-C and P-O species [24,25]. The appearance of the P-C bond indicated the successful doping of phosphorus onto the graphene sheet. The peak of the P-O bond is due to the surface oxidation of Co_2_P after air exposure. It should be noted that the P 2p_3/2_ shows a negative shift of 0.6 eV compared with the elemental P (130.1 eV) [26]. While, compared with the energy level of the zero valent state of the metal cobalt (778 eV) [23], the binding energy of Co 2p_1/2_ in the composite shows a slightly positive shift of 0.3 eV, indicating the transfer of electrons from cobalt to P. All of the above characterizations imply the successful fabrication of Co_2_P/NPG.

Hydrogen evolution reaction activity was detected through using linear sweep voltammetry (LSV) with a scan rate of 5 mV s^−1^ in 0.5 M H_2_SO_4_. For comparison, commercial Pt/C (20%), Co_2_P, and NPG were also detected. As depicted in Figure 5a, the commercial Pt/C (20%) shows the best activity, with an overpotential of 34 mV, yielding a current density of 10 mA cm^−2^. The obtained Co_2_P/NPG exhibits an overpotential of 144 mV, which is much lower than that of Co_2_P (266 mV) and NPG (467 mV). The Co_2_P/NPG exhibits superior activity than many of the recently reported non-noble metal electrocatalysts listed in Appendix A. In order to optimize the synthetic conditions, catalysts with different contents of cobalt acetate (50 mg and 200 mg) were also synthesized (labeled as Co_2_P/NPG-2 and Co_2_P/NPG-3). As shown in Appendix A, Co_2_P/NPG shows the best activity with the lowest overpotential. In addition, the HER reaction kinetic activities were estimated via Tafel slope (Figure 5b). The Tafel slope of Co_2_P/NPG is about 72 mV dec^−1^, indicating that the reaction kinetics of Co_2_P/NPG follows the Volmer–Heyrovsky mechanism. For comparison, the Tafel slope of the 20% Pt/C was also tested (29 mV dec^−1^), which is consistent with the results reported in previous publications [27,28]. It can be observed that the Tafel slope for Co_2_P/NPG is drastically lower than that of Co_2_P (114 mV dec^−1^) and NPG (196 mV dec^−1^). A lower Tafel slope typically suggests more favorable HER kinetics, further demonstrating the synergistic contribution of the Co_2_P and NPG.

Electrocatalytic stability is a key factor for evaluating the performance of the electrocatalysts, especially for further utilization in practical applications. Continuous cyclic voltammetry (CV) sweeps were conducted at a sweep rate of 100 mV s^−1^. As shown in Figure 5c, after continuous detection for 1000 cycles, the polarization curves were almost consistent with the initial test, indicating the excellent stability of Co_2_P/NPG. Moreover, a TEM image of the Co_2_P/NPG after the stability test was also test. As shown in Appendix A, after the stability test, the morphology of the sample does not show obvious changes, and the nanoparticles are evenly dispersed on the graphene sheet, further confirming the stability of the catalysts.

## 4. Conclusions

In summary, a green strategy for the preparation of N, P co-doped graphene-supported Co_2_P nanoparticles has been proposed in this study. In this strategy, the purpose of DNA is three-fold: (1) providing loading sites for catching Co^2+^; (2) acting as a green phosphorus source for the preparation of Co_2_P; and (3) acting as a N and P source for N, P doping in graphene. Due to the synergistic effect of NPG and Co_2_P, the as-prepared Co_2_P/NPG exhibits high activity, with a low overpotential (144 mV) and a small Tafel slope (72 mV dec^−1^). This study provides a successful green synthesis strategy for the preparation of high-performance TMPs.

## Figures and Tables

**Figure 1 materials-16-06119-f001:**
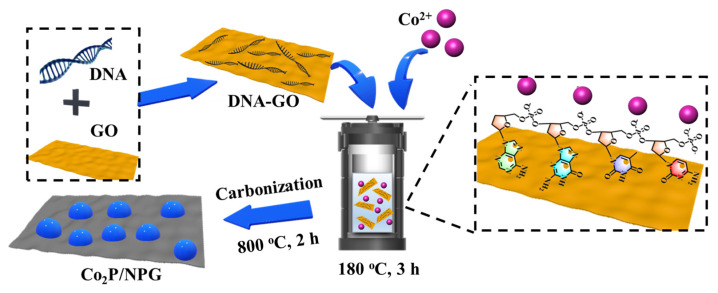
Schematic illustration of the synthesis of the Co_2_P/NPG composite.

**Figure 2 materials-16-06119-f002:**
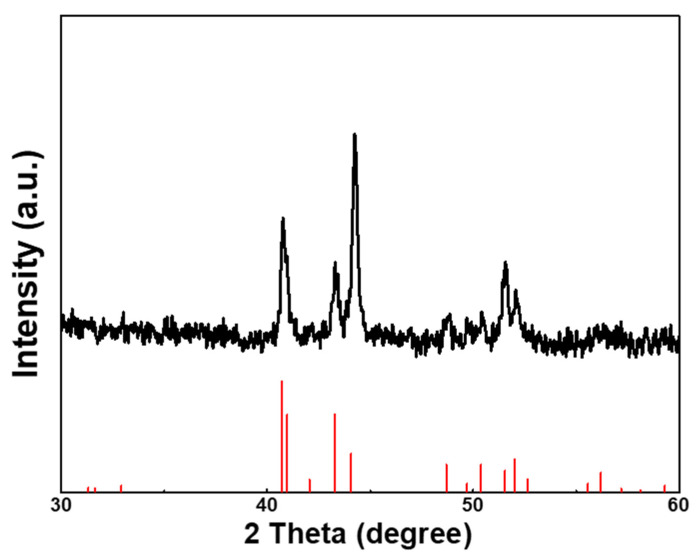
XRD pattern of Co_2_P/NPG. The vertical lines at the base of the XRD pattern represent the standard diffractions of Co_2_P.

**Figure 3 materials-16-06119-f003:**
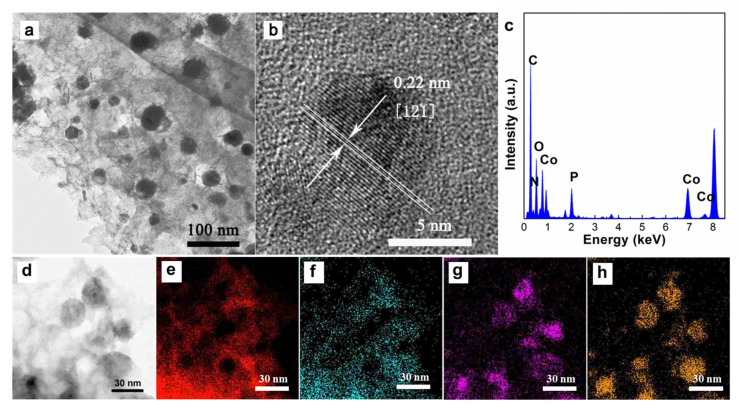
TEM (**a**) and HRTEM (**b**) images of Co_2_P/NPG. (**c**) EDS spectrum of Co_2_P/NPG. HAADF-STEM (**d**) image and the corresponding EDS mappings of C (**e**), N (**f**), Co (**g**), and P (**h**) elements for Co_2_P/NPG.

**Figure 4 materials-16-06119-f004:**
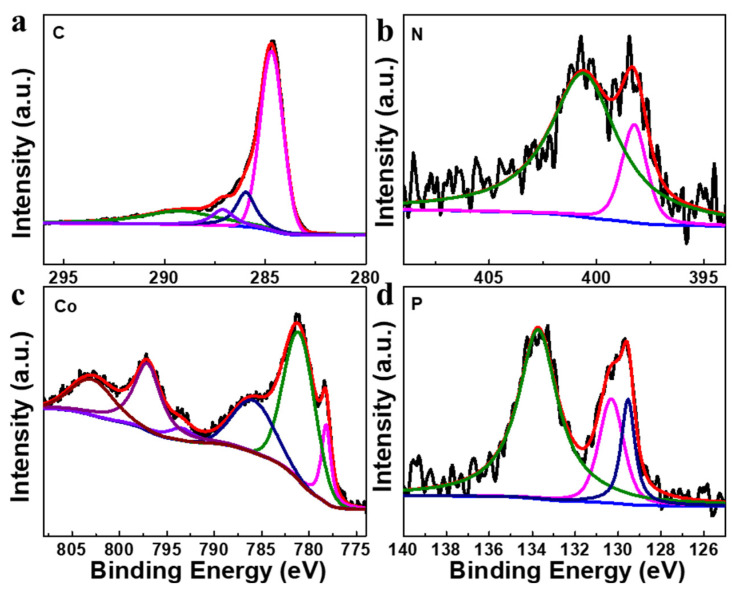
High-resolution XPS spectra of C 1s (**a**), N 1s (**b**), P 2p (**c**), and Co 2p (**d**) for Co_2_P/NPG.

**Figure 5 materials-16-06119-f005:**
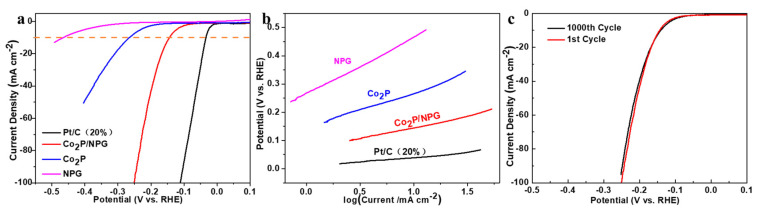
LSV curves (**a**) and corresponding Tafel plots (**b**) of Co_2_P/NPG, commercial Pt/C, Co_2_P, and NPG. (**c**) LSV curves of the Co_2_P/NPG before and after 1000 CV cycles.

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
