# Peer review of "A Green Synthesis Strategy for Cobalt Phosphide Deposited on N, P Co-Doped Graphene for Efficient Hydrogen Evolution"

_materials, 2023, doi:10.3390/ma16186119_

Round 1
Reviewer 1 Report (Previous Reviewer 2)
In the review of research article titled: A green synthesizing strategy for cobalt phosphide deposited 2 on N, P co-doped graphene for efficient hydrogen evolution by Ma et al., authors have made the good revision overall.
1. Why in XRD pattern there is no existence of substrate?
2. What was the reason of using H2SO4 for electrochemical measurements, mostly researchers use Na2SO4 or K2SO4.
3. There is much progress going on around the word about phosphates and phosphonates like in the article. “Metal Phosphates/Phosphonates for Supercapacitor Applications. In Metal Phosphates and Phosphonates: Fundamental to Advanced Emerging Applications (pp. 245-266). Cham: Springer International Publishing.”
English is good.
Author Response
- Why in XRD pattern there is no existence of substrate?
Answer: Thank you for your comment. The substrate is graphene, of which the characteristic peak is at 20~30o. The XRD spectrum in the initial manuscript is from 30~60 o. So according to your comment, we have enlarged the X axis range of the XRD image. As shown in Figure R1, we can find that there is no obvious peak between 20 and 30o, which may be due to the lower intensity of graphene comparing with that of Co2P.
- What was the reason of using H2SO4 for electrochemical measurements, mostly researchers use Na2SO4 or K2SO4.
Answer: Thank you very much for your comment. As we know, the operating condition of proton exchange membrane electrolytic cell is acidic electrolyte, so we utilize H2SO4.
- There is much progress going on around the word about phosphates and phosphonates like in the article. “Metal Phosphates/Phosphonates for Supercapacitor Applications. In Metal Phosphates and Phosphonates: Fundamental to Advanced Emerging Applications (pp. 245-266). Cham: Springer International Publishing.”
Answer: Thank you very much for your comment. I have tried my best, but I still cannot find the article “Metal Phosphates/Phosphonates for Supercapacitor Applications. In Metal Phosphates and Phosphonates: Fundamental to Advanced Emerging Applications (pp. 245-266). Cham: Springer International Publishing.” May I trouble you for giving me some guidance? Thank you again!

Reviewer 2 Report (Previous Reviewer 3)
Dear authors,
Good job with the changes.
Things that still can be improved: the authors should do a deeper characterization of the catalysts after the measurements (xrd, xps analysis...)
Author Response
Thank you for your comment!
Reviewer 3 Report (New Reviewer)
the author’s present design work, entitled is A green synthesizing strategy for cobalt phosphide deposited on N, P co-doped graphene for efficient hydrogen evolution, interesting work however few quarried need to be addressed following the below comments
1. In Figure 1 the authors should mention reaction conditions which are reaction temperature and time etc detailed schematic.
2. Figure 2 x-axis -required to change the label from 2 thetas (degree) to 2θ (degree)?
3. the authors should measure the XRD analysis for another catalyst? (Co2p/np nanoparticles molar ratio)
4. Figure 3 elemental mapping required modification, mention scaling bar (figure f-h)
5. Figure 4 required to change x-axis labeling missing (a&b)?
6. the authors can carry out a few more parametric studies such as stability, scan rate, and different ratio of the NPG malar ratio deposition on Co2p
7. The authors cite appropriate places in the following articles https://doi.org/10.1016/j.gee.2022.04.009
Author Response
Thank you for your comment. The attached file is the detailed response for the comments.

This manuscript is a resubmission of an earlier submission. The following is a list of the peer review reports and author responses from that submission.
Round 1
Reviewer 1 Report
This paper introduces that the synthesis of N, P co-doped graphene supported Co2P (CO2P/NPG) nanoparticles and their electrocatalytic properties toward HER.
Instead of a conventional method of releasing toxic substances for the synthesis of nanoparticles, the author proposed the Green synthesis strategy by using DNA. The structure of the electrocatalyst was characterized by analysis methods such as XRD, TEM, EDS, XPS, etc., and the electrochemical properties of the catalyst were investigated by methods such as LSV. Tafel slope and stability test results are also suggested.
The idea and direction of the paper are good, but the following will need to be supplemented for publication. Especially, it must have a format suitable for the publication.
Comments
1. Characterization is not enough to prove the structure of catalyst.
The synthesis process of the scheme (Figure 1) is too notional. What happens to DNA after synthesis? What is the change in GO? There has not been a sufficient explanation for how it changes after carbonization.
It is natural that the peak of the element put in comes out at the XPS data.
In the XRD data, the picture resolution of figure 2a is low, making it difficult to perform peak assignment. It is difficult to be sure that the product is really Co2P, a reference material.
2. Actually discussion for the experimental data is required.
3. English polishing and manuscript editing is required: What are the “NPG” and “TMPs” in abstract? They are used without explanation of the full name. Many typos : Co2P , PH3, Figure 3 (missing period), and etc.
Figure 3c and 3d are switched according to the figure captions. Y-axis scale is required.
Extensive editing of English language required
Author Response
Thak you very much for your comment. You can find our point-by-point response in the attached file.

Reviewer 2 Report
In the review of the manuscript titled: “A green synthesizing strategy for cobalt phosphide deposited on N, P co-doped graphene for efficient hydrogen evolution” by Ma et al., authors have presented the good work, capable to be published in the journal after minor modifications. The suggestions are as follow;
1. Abstract is well written, precise and to the point, there should be a sentence about the electrochemical measurements with their values.
2. Introduction section should contain few updated citations like in the paragraph 2, it should be like “Various non-noble metal electrocatalysts in the form of transition metal oxides (MnO2, Bi2O3, SnO2 etc), transition metal sulfides (Bi2S3, Co3S4, etc) and transition metal phosphides (CoP, NiP etc) [Fundamental to Advanced Emerging Applications (pp. 245-266). Cham: Springer International Publishing.] have presented superior catalytic performance in hydrogen evolution reaction”.
3. In the XPS analysis please calculate the ΔE value for Co and P and compare your result with the standard literature.
4. Very little explanation of CV (Figure 4) is there not enough to explain the phenomenon, please elaborate it.
English of the manuscript is good, a little spelling or grammar should be re-checked.
Author Response

(The authors gave the same response as above.)

Reviewer 3 Report
Dear authors,
There are some point that can be addressed to improve the manuscript:
1) Some subscripts are missing. I have highlighted in the text. Please, correct them.
2) You should define the acronysms before you use them in the text.
3) In the results section, the first paragraph has to be remove.
4) The morphology and composition of Co2P/NPG catalyst after durability test should be investigated to demonstrate its stability.
5) In the discussion section a paragraph and/or a table should be include about other catalyst in order to compare their performance with the Co2P/NPG catalyst

Author Response

(The authors gave the same response as above.)

Round 2
Reviewer 1 Report
no
no